# The Effect of Microwave on the Crystallization Behavior of CMAS System Glass-Ceramics

**DOI:** 10.3390/ma13204555

**Published:** 2020-10-14

**Authors:** Hangren Li, Saiyu Liu, Wence Xu, Yuxuan Zhang, Xin Li, Shunli Ouyang, Guangkai Zhao, Fang Liu, Nannan Wu

**Affiliations:** 1Key Laboratory of Integrated Exploitation of Bayan Obo Multi-Metal Resources, Inner Mongolia University of Science and Technology, Baotou 014010, China; Li_hangren@outlook.com (H.L.); 2018023227@stu.imust.edu.cn (S.L.); xumence910707@163.com (W.X.); ldu2013yuxuan@outlook.com (Y.Z.); leexinche@gmail.com (X.L.); 2019022059@stu.imust.edu.cn (G.Z.); liufangbaotou2008@163.com (F.L.); woshinannan04@imust.cn (N.W.); 2Open Project of State Key Laboratory of Advanced Special Steel, Shanghai Key Laboratory of Advanced Ferrometallurgy, Shanghai University, Shanghai 200444, China

**Keywords:** microwave sintering, glass-ceramics, non-thermal microwave effect, Raman spectroscopy

## Abstract

The microwave sintering of glass-ceramics, non-thermal microwave effect, and crystal growth mechanism remain important challenges in materials science. In this study, we focus on developing approaches to affect crystal growth in the glass network of glass-ceramics by microwave heating, rather than performing a single study on the crystal structure and type. Raman spectroscopy is used to detect the structure of the glass network. We demonstrated that the non-thermal microwave effect promoted the diffusion of metal ions, which promoted the aggregation and precipitation of metal ions in the glass network to form crystals. The samples produced by microwave heating contain more non-bridging oxygen bonds than conventional sintered samples; therefore, the non-thermal microwave effect has a depolymerization effect on the glass network of the sample. Under the influence of microwave field, many metal ions precipitate, which precipitates many crystal nuclei. In addition, many active metal ions are captured during the crystal nucleus growth, which shortens the sintering process of glass-ceramics.

## 1. Introduction

In the 1960s, microwaves were used to sinter materials [1]. Many studies have shown that microwave sintering can greatly shorten the sample preparation process and accelerate crystal growth. This occurs mainly owing to non-thermal and thermal microwave effects [2,3,4]. The thermal microwave effect is mainly manifested in differences in thermodynamics, including problems such as faster heating rates and local hot spots [5,6]. The essence of microwave sintering is that microwave field drives the movement of charged particles and dipoles in the material to generate heat, while conventional sintering mainly relies on the combination of external heat radiation and convection heat conduction to absorb heat [7,8,9]. Non-thermal microwave effect refers to the difference in materials produced by different heating principles of the microwave sintering and conventional sintering under the same thermodynamic conditions. Different from conventional furnace heating, the main characteristic of microwave heating is that the electromagnetic wave in the microwave field penetrates most of the dielectric materials, and the body is heated by integral absorption [10,11]. The main mechanism of absorption is through the excitation of electronic oscillations at the microwave source frequency. The non-thermal microwave effect reduces the activation energy during the sintering process and enhances the movement of charged particles, which result in the rapid formation of the crystal nucleus and rapid growth of crystal [2,12].

Glass-ceramics is an environmentally friendly new material that consists of crystalline and amorphous structural phases [13]. This material has the advantages of low cost, acid and alkali resistance, and good mechanical properties [14]. Besides, it can effectively deal with solid waste and use it as a resource. In the past, research on the rapid growth of crystals and enhancement of structure and performance was carried out in the studies on producing glass-ceramics by microwave heating [15,16,17,18,19]. However, there are only few studies on amorphous state. Mahmoud [20], during the research on lithium disilicate glass-ceramics by microwave heating, characterized the glass network by Raman spectroscopy. He determined that the microwave field promoted the transformation of Q^3^ to Q^2^ and Q^4^ during nucleation, and crystallization was developed. Li [21,22] has researched the microwave sintering of CaO-MgO-Al_2_O_3_-SiO_2_ (CMAS) glass-ceramics by Raman spectroscopy. He determined that with an increase in heat treatment temperature, the characteristic band strength of each sample gradually increased. This result indicated that the amount of diopside crystal precipitation gradually increased, and the degree of order, compactness, and integrity of diopside crystal structure increased. Raman spectroscopy is one of the best research tools for glass networks [23]. There are many studies on glass-ceramics by conventional heating, which used Raman spectroscopy [24,25,26,27,28,29]. In the studies on CMAS glass-ceramics by conventional heating, during the process of nucleation heat treatment, ion migration and phase separation are induced in glass by ions created by high field strength. Thus, the glass network is damaged. It is determined that Q^4^ with four bridging oxygen bonds and Q^2^ with two bridging oxygen bonds are transformed into Q^3^ with three bridging oxygen bonds and Q^1^ with one bridging oxygen bond. The polymerization degree of the entire glass network is reduced, and the heterogeneous nucleation point is formed to induce crystallization [28]. These studies are based on the analysis of Raman spectroscopy.

Raman spectroscopy can accurately show changes in the silicon–oxygen network and alkali metal ions during the preparation of glass-ceramics. By analyzing these changes, the nucleation mechanism of glass-ceramics and the role of basic metal ions in the preparation process can be determined. Amorphous research is mainly focused on glass networks. In this work, Raman spectroscopy is used as the main tool to study the effect of microwaves on the growth of crystals, research the effect of non-thermal microwaves on glass networks and alkali metal ions, and to explore their association in promoting crystallization.

## 2. Experimental

### 2.1. Sample Preparation

According to the phase diagram of the CMAS system and relevant documents, the basic formula of the CMAS system glass-ceramics is designed using the raw materials, as shown in Table 1. Cr_2_O_3_ and Fe_2_O_3_ is a nucleating agent, which is added in the form of stainless-steel slag. Together with Cr_2_O_3_, Fe_2_O_3_ acts as a composite nucleating agent to promote the overall crystallization of glass-ceramics. By adjusting the amount of ferrochrome slag introduced, the content ratio of the composite nucleating agent is changed to increase the ratio of the nucleating agent in the raw material. The rest of raw materials are chemically pure.

According to the formula of basic glass, the required amounts of materials are calculated and weighed according to the determined proportion. After milling and mixing, the material is put into an alumina crucible and heated to 1500 °C in a resistance furnace for 3 h for melting and clarification. Then, a part of glass melt was quenched, dried, and powdered for differential scanning calorimetry (DSC) measurements. The remaining glass melt was poured into a pre-heated stainless-steel mold (40 × 60 × 8 mm), demolded after molding, and the sample was put into a 600 °C muffle furnace for 5 h for annealing.

After annealing, the samples were placed in a muffle furnace (Yunjie, Baotou, China) and a 2.45 GHz multi-mode cavity microwave oven (Hunan Huae Microwave Technology Co., Ltd., Changsha, China). Silicon carbide was used as the auxiliary heating medium to conduct the microwave crystallization heat treatment according to the heat treatment system shown in Table 2. Microcrystalline glass samples were obtained with the numbers of C-1, C-2, C-3, M-1, M-2, and M-3. The microwave oven used in the experiment is an HY-MF3016-type multi-mode resonant cavity microwave heater (Hunan Huae Microwave Technology Co., Ltd., Changsha, China) with adjustable power of 0–3 kW and working frequency of 2450 MHz. The temperature measurement system uses a k-level thermocouple to directly measure the sample temperature.

### 2.2. Sample Characterization

The crystalline phase was determined by X-ray diffraction (XRD, X’pert Pro Powder PANalytical B. V., Almelo, The Netherlands) using Cu K α radiation with a 0.1541 nm wavelength in the 2-theta/° range of 10–80°, step size of 0.02°, and scan speed of 0.3 s/min at room temperature. The instrument was operated at 40 kV and 40 mA. The XRD data were fitted using X’Pert HighScore Plus (Version: 3.0, Almelo, The Netherlands), and the degree of crystallinity of the specimens was obtained by the “Constant Background” value. The microstructure was evaluated by a field emission scanning electron microscopy (FESEM, SUPRA 55 FESEM) instrument (Carl Zeiss, Jena, Freistaat Thüringen, Germany), which was equipped with an Oxford energy dispersive spectrometer (EDS) analysis system. The specimens were polished, corroded, and coated with gold. The Raman spectroscopy system was constructed independently by our laboratory, equipped with a Renishaw inVia Raman system. In the experiment, a 50× long focus objective lens (parameters: 50×/0.35) was used for focusing on the specimens. Focusing was achieved by adjusting the specimen height, and each specimen was excited with a 532 nm wavelength semiconductor laser (Renishaw, London, UK). Then, the Raman spectrometer was calibrated with a single crystal silicon prototype. All signal data of each specimen were collected at room temperature with 5 s exposure time and 3 accumulations. The scanning range was from 200 to 1900 cm^−1^. The hardness of the samples was measured by the Vickers hardness method. The three-point flexural strength (FS) of the rectified parallelepiped bars (3 × 4× 40 mm) of the sample was tested using the CSS-88000 electronic universal testing machine (Changchun testing machine research institute, Changchun, China).

## 3. Results and Discussion

### 3.1. Thermal Analysis

Figure 1 shows the DSC curve of the base glass. In the curves of all three samples, there are clear sharp exothermic peaks at 959, 894, and 874 °C. As shown in Figure 1, as the content of the composite crystal nucleating agent increases, the glass crystallization peak gradually shifts to the low-temperature direction, and it can be seen that the glass crystallization temperature gradually decreases. At the same time, the glass transition temperature (T_g_) is not very obvious, and no obvious exothermic peak of glass nucleation is found in Figure 1. We used a two-step heat treatment system to prepare glass-ceramics. The glass nucleation temperature is 30–50 °C higher than the glass transition temperature. On the basis of DSC results, the heat treatment system is determined, which provides a reference for the preparation of glass-ceramics by conventional and microwave heating. To study the effect of non-thermal microwaves on the sintering process of glass-ceramics, the nucleation and crystallization temperatures of microwave-sintered samples are made to be the same as those of conventional sintered samples, and the nucleation and crystallization temperatures are 720 and 880 °C, respectively. However, owing to the special effect of microwaves, to ensure that the samples are completely crystallized, the soaking time of nucleation and crystallization time of the samples were 30 and 20 min, respectively. Conventional sintering uses heat radiation as the heat source, and sintering speed is much slower than that of microwave sintering. To ensure the complete nucleation and crystallization of conventional samples, the holding time of nucleation and crystallization stages is set to 3 h.

### 3.2. X-ray Diffraction Analysis

Figure 2 shows the XRD pattern of the prepared glass-ceramics, which was nucleated and crystallized by conventional and microwave heating. Figure 2 shows that there is only a very small crystallization peak in C-1 and M-1 samples. It can be seen that the basic glass species only contains a small amount of Fe_2_O_3_ as a nucleating agent, which fails to promote glass crystallization at 880 °C. It was determined that the crystallinity of the two samples is very low, and there is a distinct broadened amorphous peak. C-2, C-3, M-2, and M-3 samples were successfully crystallized and exhibited almost identical characteristic peaks. All crystal phases are a diopside phase (ICSD: 00-041-1370). The characteristic peak intensity of microwave samples (M-2, M-3) at 30° is stronger than that of conventional samples (C-2, C-3). This result indicates that the use of microwave heat treatment of glass can induce glass to devitrify in a shorter holding time for nucleation and crystallization. Also, the diffraction peak intensity of the traditional heat treatment sample with the same chromium oxide content in the crystallized sample is lower than the diffraction peak intensity of the microwave heat treatment sample. It can be seen that the crystallinity of microwave heat-treated samples is higher than that of traditional heat-treated samples.

### 3.3. Scanning Electron Microscopy

The microstructure and surface crystallization of the sample are shown in Figure 3. The samples produced by conventional and microwave heating without adding Cr_2_O_3_ mainly show surface devitrification. It can be seen from the figure that when no chromium oxide is added to the raw materials and only a small amount of Fe_2_O_3_ is heat-treated at 880 °C, the sample fails to achieve the purpose of crystallization, and only a slight surface crystallization occurs. The information that the sample is mainly glassy is also obtained from the XRD diffraction pattern. C-2 and M-2 samples have a similar dendrite structure. The grain size of the microwave heat-treated samples is refined, and the structure is relatively uniform. C-3 and M-3 samples have the same globular crystal and dense structure, and the microwave sample has a larger grain size. At the same sintering temperature, the samples with shorter sintering time have denser microstructure and larger grains. This result shows that the crystallization kinetics of glass-ceramics during nucleation and crystallization is enhanced by the non-thermal microwave effect. By promoting the diffusion of metal ions and reducing the crystallization activation energy, the damage of the glass network is intensified, and the precipitation and growth of crystal nucleus are accelerated.

The EDS analysis of two samples was performed, and the results are shown in Figure 4. By comparing EDS results between C-2 and M-2 samples, it was determined that the amount of Fe and Cr in the microwave sample is higher. The same trend was observed in C-3 and M-3 samples. According to the characteristics of microwave heating, microwaves can selectively couple with oxides in glass. Both iron oxide and chromium oxide have a strong ability to absorb microwaves, and preferentially react and migrate and agglomerate during the process of crystallization and heat treatment of glass by microwave. Therefore, under the action of microwave, iron oxide and chromium oxide play the role of nucleating agents to form agglomerates to promote the crystallizing growth of the epitaxial iron oxide and chromium oxide in the glass phase.

### 3.4. Raman Spectroscopy

There are two types of oxygen in pyroxene silicon tetrahedral, i.e., bridging oxygen (BO) and non-bridging oxygen (NBO). Bridging oxygen is coordinated by two Si-O networks, and the bond between bridging oxygen and other cations are very weak, while non-bridging oxygen bonds connect more cations [30]. Therefore, the mechanical constants of Si-O^0^ and Si-O^−^ are different. The silicon oxygen bond length of bridging oxygen is longer than that of non-bridging oxygen bonds; therefore, the wave number of bridging oxygen expansion vibration absorption is lower than that of non-bridging oxygen bond expansion vibration absorption [31,32,33]. In the wave number range of 800–1200 cm^−1^, pyroxene exhibits a different Raman spectrum of silica tetrahedron (Q^n^). The Raman bands at 1100–1200 cm^−1^, 1050–1100 cm^−1^, 950–1000cm^−1^, 900 cm^−1^, and 850 cm^−1^ show Si-O stretching vibration Raman peaks with 0 non-bridging oxygen bonds of silica tetrahedron (SiO_4_) (Q^4^), 1 non-bridging oxygen bond of silica tetrahedron (Si_2_O_5_)^2−^ (Q^3^), 2 non-bridging oxygen bonds of silica tetrahedra (SiO_3_)^2−^ (Q^2^), 3 non-bridging oxygen bonds of silicon oxygen tetrahedra (Si_2_O_7_)^6−^ (Q_1_), and 4 non-bridging oxygen bonds of silicon oxygen tetrahedra (SiO_4_)^4−^ (Q^0^), respectively.

Figure 5 shows the Raman spectrum of annealed glass and glass-ceramics samples. In Figure 3, the Raman peak at 300–600 cm^−1^ is due to Si-O bending vibration, and the Raman peak at 800–1200 cm^−1^ is due to Si-O stretching vibration. It is observed that conventional (C-1) and microwave (M-1) samples are only manifested as the glass phase owing to the absence of added Cr_2_O_3_ in the S1 sample. This result is similar to the Raman peak of annealed glass (G-1). By analyzing the Raman peak at 800–1200 cm^−1^, it is determined that the full width at half maximum (FWHM) of the sample decreases in the order of FWHM_G-1_ > FWHM_C-1_ > FWHM_M-1_. Owing to the addition of Cr_2_O_3_ to S2 and S3 samples, there are sharp crystal peaks near 330, 400, 675, and 1000 cm^−1^ of glass-ceramics. The Raman peaks of conventional samples are similar to those of microwave samples. For the Raman peak at 800–1200 cm^−1^, the same decreasing trend is observed as that for the S1 sample (FWHM_G-2_ > FWHM_C-2_ > FWHMM-2; FWHM_G-3_ > FWHM_C-3_ > FWHM_M-3_). A decrease in the corresponding FWHM indicates that the distribution width of the structural elements of Q^n^ becomes narrow. This occurs owing to the network accumulation effect of the larger cation field strength of Cr and other elements in the glass network structure.

After resolving and fitting the Raman peak at 800–1200 cm^−1^, the fitting results are shown in Figure 5 and Table 3. The Raman peak at 800–1200 cm^−1^ of annealed glass and C-1 and M-1 samples is divided into four Raman peaks, including Q^1^, Q^2^, Q^3^, and Q^4^, while those of C-2, M-2, C-3, and M-3 samples are divided into six Raman peaks, including Q^0^, Q^1^, Q^2^, Q^2^(N), Q^3^, and Q^4^, in which Q^2^(N) is the chain structure of (SiO_3_)^2−^ in diopside crystal. By analyzing the peak position for samples S1, S2, and S3, it is determined that the peak position of glass-ceramics samples is red-shifted relative to annealed glass samples. The peak position of glass-ceramics samples produced by microwave heating is red-shifted compared with that of glass-ceramics samples obtained by conventional heating.
(1)$$\nu \;=\;\frac{1}{2\pi c}\;\;\sqrt{\frac{k}{\mu }}\;\;\;$$
where μ is the reduced mass, C is the speed of light, k is the force constant of Si-NBO bonds, and ν is the Raman band peak position. The k value is related to metal ions in the glass network, e.g., Cr and Fe. The Cr ion has a larger cation field strength; thus, the Cr ion can aggregate with other metal oxides (e.g., MgO, FeO, and Fe_2_O_3_) in the glass network and precipitate in the form of a spinel crystal [34]. Fe^2+^ in silicate network exists in the form of octahedral coordination, which plays a role in the breaking of the glass network; thus, it can reduce glass viscosity, form another phase that is different from the glass structure produced during heat treatment, and promote the nucleation and crystallization of glass. Fe^3+^ in the silicate network can exist in the form of tetrahedral coordination and octahedral coordination with a complex mechanism. In the form of tetrahedral coordination, which is the mechanism for Al^3+^, in the case of compensating charge of monovalent alkali metal ions and divalent alkali earth metal ions, tetrahedral iron oxide (FeO_4_) enters into the glass network structure. In the presence of octahedral coordination, the mechanism is the same as that of Fe^2+^, which can destroy the network and reduce viscosity. For glass-ceramics, the nucleation and crystallization of glass-ceramics is a process in which metal ions gather and precipitate to form a crystal nucleus. During this process, metal ions with high field strength (e.g., Cr and Fe) precipitate first, which leads to the corresponding increase in k value and reduction of mass, and finally to an increase in ν. The non-thermal microwave effect can enhance the diffusion of metal ions and promote more metal ions to gather and then precipitate. Thus, the ν of glass-ceramics samples is larger than that of annealed glass samples, and the ν of microwave samples is larger than that of conventional samples.

The fitting results are calculated and analyzed using the theory proposed by James E. Shelby and the method of Norimasa Umesaki, which are used to research alkali silicate glass and cylinder structure [35,36], as shown in Table 4. Among them, X_i_ is the mole fraction of (SiO_4_) tetrahedral structural units with non-bridging oxygen bonds. X_i_ is related to Ai%, as follows: X_i_ = a_i_A_i_%. The a_i_ is the normalized Raman cross-section of (SiO_4_) tetrahedral structural units. The corresponding ai values of Q^1^, Q^2^, and Q^3^ are, respectively, 1.15, 1.02, and 1.04. Although there is no a_4_, X_4_ can be calculated by X_4_ = 1 − $\sum X_{i}$. In addition, Q_n_ = ([Si] + [Al]) × X_i_.

Several important parameters are shown as follows:(2a)$$\frac{\mathrm{NBO}}{\mathrm{NB}0\;+\;\mathrm{BO}}=\frac{\sum Q^{n}\;\times \;a}{\;\left[O\right]}\;\;$$
(2b)$$\frac{\mathrm{NBO}}{\mathrm{Tetrahedron}}=\frac{\;\sum \{Q^{n}\times \left(4-n\right)\}}{\left[\mathrm{Si}\right]\;+\;\left[\mathrm{Al}\right]}\;\;\;$$
(2c)$$\frac{\mathrm{BO}}{\mathrm{Tetrahedron}}=\frac{\sum \{Q^{n}\times n\}}{\;\;\left[\mathrm{Si}\right]+\;\left[\mathrm{Al}\right]}\;\;\;\;$$
where [O] is the total molar concentration of oxygen for basic glasses, and [Si] + [Al] is the molar concentration of tetrahedral structural units. Table 4 shows that the molar content of Q^2^ and Q^1^ gradually increases with the addition of Cr_2_O_3_, which means that the failure degree of the glass network increases. Moreover, the failure degree of samples produced by microwave heating is more severe than that of samples produced by conventional heating. The fraction of non-bridging oxygen, the average number of NBO per tetrahedron, and the average number of bridging corners per tetrahedron is obtained by quantitative calculation. By comparing the number of bridging and non-bridging oxygen bonds, it is determined that the number of bridging oxygen bonds in microwave samples is considerably lower than that in conventional samples, and the number of non-bridging oxygen bonds is higher than that in conventional samples, which shows that the non-thermal microwave effect has a depolymerization effect on the glass network.

Figure 6 shows a comparison diagram of NBO/tetrahedron and BO/tetrahedron between annealed glass and glass-ceramics produced by conventional and microwave heating. With the addition of steel slag, the content of Cr_2_O_3_ increases. Owing to its high field strength, Cr_2_O_3_ precipitates with other metal oxides in the glass mesh, which results in the depolymerization of the glass network. Meanwhile, microwave-sintered samples exhibit a depolymerization effect on the glass network compared with conventionally sintered samples. However, when the amount of chromium is too large, owing to the high field strength of chromium, there is an aggregation effect on the glass network; thus, the S3 sample shows that the aggregation degree of the glass network increases.

### 3.5. Vickers Hardness

The Vickers hardness test results of the samples are shown in Table 5. Hardness is related to the microstructure of the sample. Sample hardness increases with a decrease in the grain size. Table 5 shows that the C-3 sample has the highest hardness. The grain size of the C-3 sample is the smallest, and the sample has a compact microstructure. The M-1 sample contains fine crystals; therefore, M-1 has higher hardness than C-1. The hardness of C-2 and M-2 samples is similar owing to their similar microstructure.

## 4. Conclusions

In this paper, chromium-containing waste slag was used as the main raw material to prepare slag glass-ceramics using traditional heat treatment methods and microwave heat treatment. Both heat treatment methods can successfully prepare pyroxene phase glass-ceramics. The experimental results show that the microwave method can promote pyroxene phase crystallization in a shorter time and at a lower temperature. The non-thermal effects of microwave were studied by Raman analysis and testing technology. Meanwhile, the peak position of Q^1^–Q^4^ in microwave samples blue-shifts compared with that of conventional samples. Microwave samples are prone to form a glass network, and the non-thermal microwave effect depolymerizes the glass network. The microstructure of microwave samples is more compact, and the grains are larger than those of conventional samples. Owing to the non-thermal microwave effect on the enhanced diffusion of Fe^3+^, many fine nuclei appear in the M-1 sample without adding Cr_2_O_3_, and the surface crystallization layer is much wider than that in the C-1 sample. Thus, the rapid sintering speed and long holding time used in this experiment produce grain sizes that are too large, and the hardness of the sample is not very ideal. This research provides a theoretical basis for the rapid nucleation and crystallization during the process of non-thermal microwave heating. Thus, this study promotes the preparation of glass-ceramics and provides a new approach for the preparation of glass-ceramics by microwave heating.

## Figures and Tables

**Figure 1 materials-13-04555-f001:**
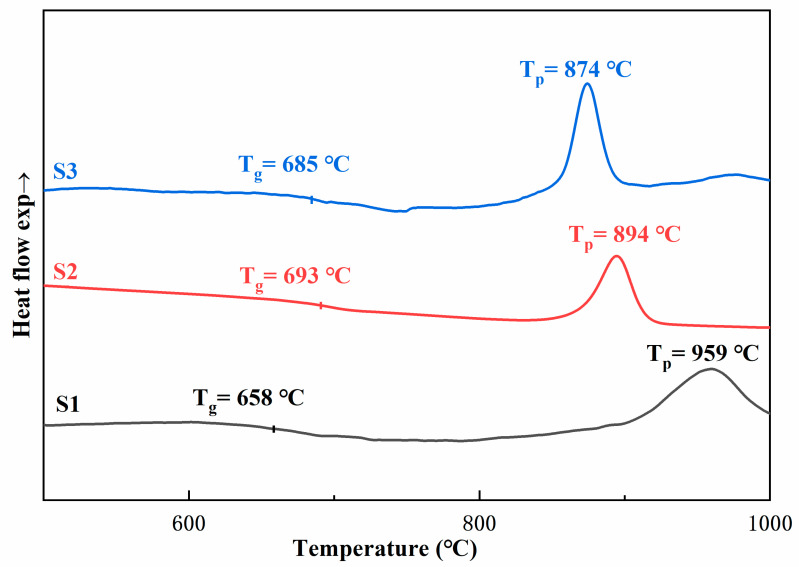
Differential scanning calorimetry (DSC) curve recorded for the glass matrix.

**Figure 2 materials-13-04555-f002:**
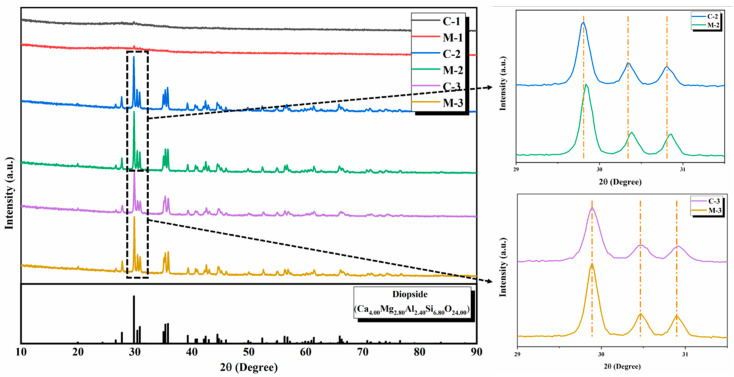
X-ray diffraction (XRD) pattern of crystallized glass-ceramics by conventional and microwave heating.

**Figure 3 materials-13-04555-f003:**
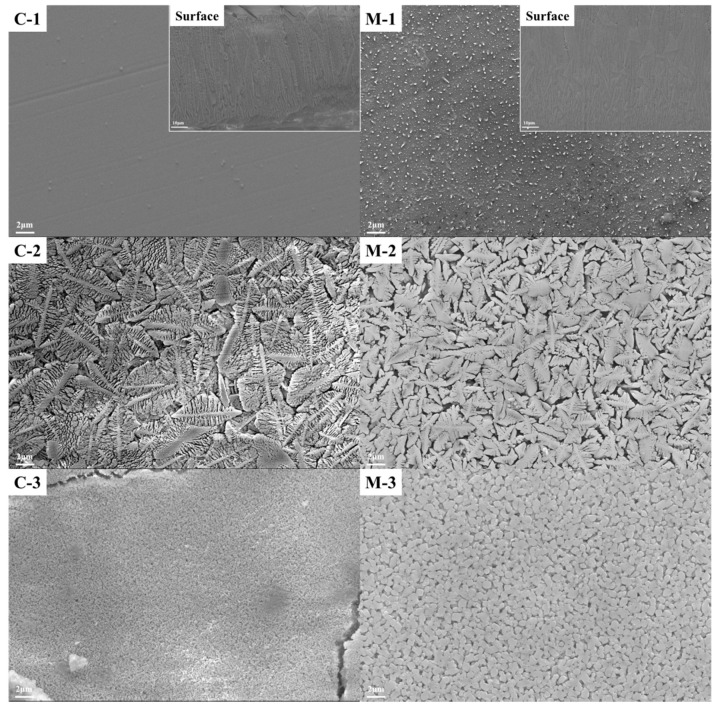
Scanning electron microscopy (SEM) images of glass-ceramics produced by conventional (C-n) and microwave (M-n) heating.

**Figure 4 materials-13-04555-f004:**
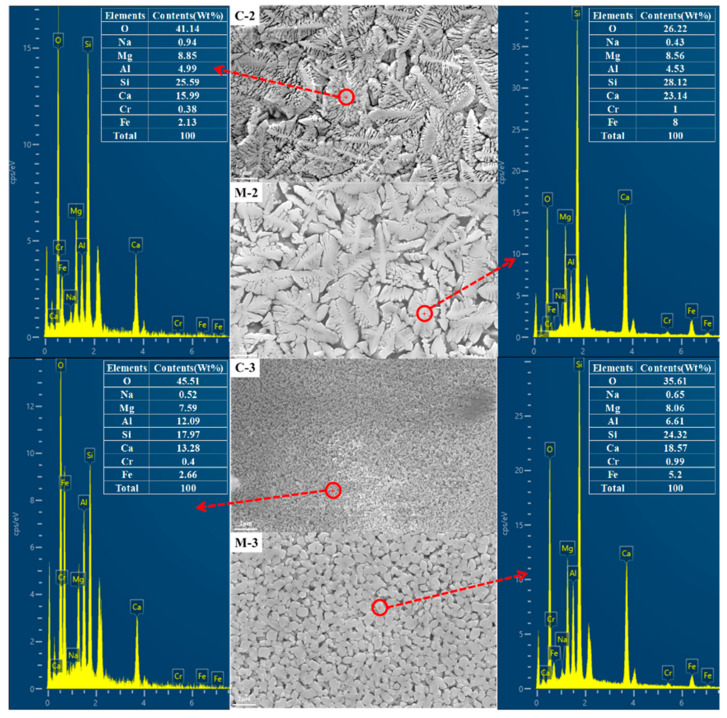
SEM image and EDS analysis of glass-ceramics produced by conventional (C-2, C-3) and microwave (M-2, M-3) heating.

**Figure 5 materials-13-04555-f005:**
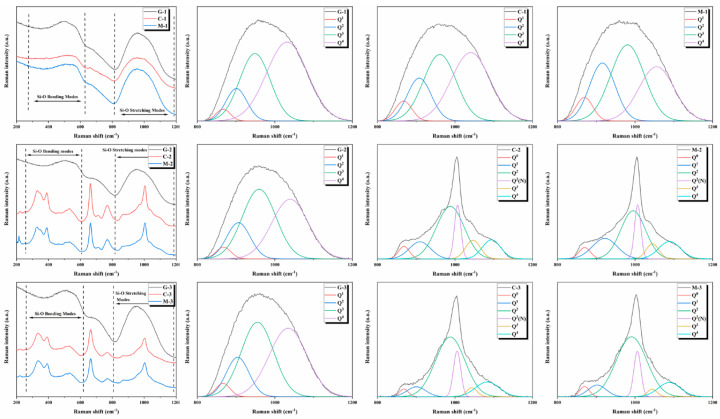
Raman spectra of annealed glass and crystallized glass-ceramics produced by conventional heating and microwave heating.

**Figure 6 materials-13-04555-f006:**
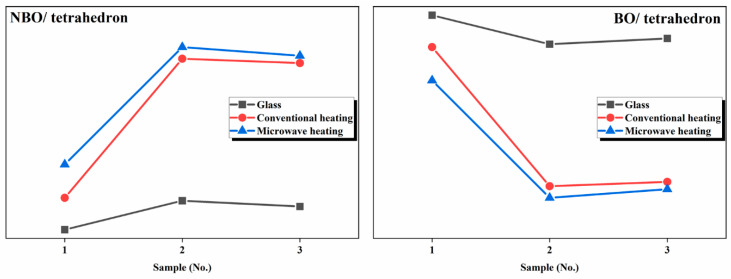
Comparison of NBO/tetrahedron and BO/tetrahedron between annealed glass and glass-ceramics by conventional heating and microwave heating.

**Table 1 materials-13-04555-t001:** Chemical composition of basic glass.

Sample	Chemical Composition (mol%)							
No.	MgO	SiO_2_	CaO	Na_2_CO_3_	Na_2_B_4_O_7_	Al_2_O_3_	Fe_2_O_3_	Cr_2_O_3_
S1	14.17	52.54	20.24	3.74	1.36	5.60	2.36	0.00
S2	14.36	52.35	20.03	3.63	1.33	5.74	2.35	0.20
S3	14.62	52.11	19.82	3.52	1.32	5.92	2.29	0.40

**Table 2 materials-13-04555-t002:** Heat treatment process of glass-ceramics samples.

Sample No.	Nucleation	Crystallization	Heat Treatment
C-1	720 °C, 3 h	880 °C, 3 h	Conventional heating
C-2	720 °C, 3 h	880 °C, 3 h	Conventional heating
C-3	720 °C, 3 h	880 °C, 3 h	Conventional heating
M-1	720 °C, 30 min	880 °C, 20 min	Microwave heating
M-2	720 °C, 30 min	880 °C, 20 min	Microwave heating
M-3	720 °C, 30 min	880 °C, 20 min	Microwave heating

**Table 3 materials-13-04555-t003:** Frequencies (ν_n_), FWHM (cm^−1^), and area% (A_n_%) of Raman bands obtained from the deconvolution fits.

Item	G-1	C-1	M-1	G-2	C-2	M-2	G-3	C-3	M-3
ν_1_	865.09	866.00	867.98	866.26	907.56	919.78	864.94	898.67	900.45
ν_2_	899.03	905.82	913.72	904.25	985.36	992.57	902.19	985.11	988.42
ν_3_	946.28	958.43	978.10	957.09	1043.84	1042.41	952.90	1040.82	1041.72
ν_4_	1029.25	1037.86	1053.23	1036.92	1091.99	1086.92	1031.95	1081.14	1086.42
FWHM_1_	42.50	50.07	50.88	44.16	63.91	74.29	42.49	56.57	54.14
FWHM_2_	63.03	72.17	77.60	70.82	83.87	75.10	68.32	89.02	92.39
FWHM_3_	92.41	98.80	101.82	101.34	45.11	38.49	97.89	32.07	29.66
FWHM_4_	125.77	118.42	107.74	111.38	56.58	66.45	115.21	72.40	68.48
A_1_	94,203	186,000	145,160	91,922	224,390	248,050	106,040	57,747	64,473
A_2_	380,570	576,550	544,290	452,600	901,900	587,490	501,990	544,510	566,400
A_3_	1,161,100	1,222,800	937,670	1,247,500	173,230	94,918	1,357,000	29,659	22,439
A_4_	1,847,300	1,508,000	707,080	1,173,300	221,080	188,510	1,469,000	110,210	99,534
A_1_%	2.70	5.32	6.22	3.10	14.76	22.17	3.09	7.78	8.56
A_2_%	10.93	16.50	23.32	15.26	59.31	52.50	14.62	73.37	75.23
A_3_%	33.33	35.00	40.17	42.07	11.39	8.48	39.52	4.00	2.98
A4%	53.03	43.17	30.29	39.57	14.54	16.85	42.78	14.85	13.22

**Table 4 materials-13-04555-t004:** Fraction (X^n^%) and content (Q^n^) of structure units, fraction of non-bridging oxygen, average number of NBO per tetrahedron, and average number of bridging corners per tetrahedron.

Item	G-1	C-1	M-1	G-2	C-2	M-2	G-3	C-3	M-3
X^1^%	3.11	6.12	7.15	3.56	16.97	25.49	3.55	8.95	9.85
X^2^%	11.14	16.83	23.78	15.57	60.50	53.55	14.91	74.84	76.74
X^3^%	34.67	36.40	41.78	43.75	11.85	8.82	41.10	4.16	3.10
X^4^%	51.08	40.64	27.29	37.11	10.68	12.13	40.44	12.06	10.31
Q^1^	1.98	3.90	4.56	2.28	10.83	16.27	2.27	5.72	6.29
Q^2^	7.10	10.73	15.16	9.94	38.61	34.18	9.53	47.82	49.04
Q^3^	22.10	23.20	26.63	27.92	7.56	5.63	26.26	2.66	1.98
Q_4_	32.56	25.90	17.39	23.69	6.82	7.74	25.84	7.70	6.59
NBO/(NBO + BO)	0.23	0.31	0.38	0.30	0.64	0.67	0.28	0.63	0.65
NBO/tetrahedron	0.66	0.88	1.11	0.86	1.84	1.92	0.82	1.81	1.86
BO/tetrahedron	3.34	3.12	2.89	3.14	2.16	2.08	3.18	2.19	2.14

**Table 5 materials-13-04555-t005:** Vickers hardness of glass-ceramics produced by conventional and microwave heating.

Sample (No.)	C-1	C-2	C-3	M-1	M-2	M-3
Vickers hardness (HV 0.5)	701.24 ± 18.05	819.79 ± 26.47	908.91 ± 32.44	739.43 ± 32.92	821.23 ± 28.69	848.04 ± 40.49

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
