# Peer review of "The Effect of Microwave on the Crystallization Behavior of CMAS System Glass-Ceramics"

_materials, 2020, doi:10.3390/ma13204555_

Round 1

Reviewer 1 Report

In this work, the crystallization processes have been analyzed for CMAS glass-ceramic samples, which were obtained using traditional heat treatment methods and microwave heat treatment. Samples were examined using different experimental techniques: DSC, XRD, SEM+EDS and Raman spectroscopy. In my opinion, the results are interesting from the structural point of view. Glass-ceramics based on the system CaO-MgO-Al2O3-SiO2 (CMAS) are important for numerous applications.

In present paper, the Authors compared structure, properties and crystallization behavior of glass-ceramics CMAS, which were fabricated using traditional heat treatment methods and microwave heat treatment. In my opinion, the manuscript is well-written and contains interesting data from the structural point o view. The results give important contribution to the glass-ceramic science and technology. I did not find any content-related mistakes. The manuscript is recommended for publication in the Materials.

Author Response

It is my pleasure to get such a high evaluation from you, and we have carefully revised and checked the spelling and expression of English

Reviewer 2 Report

The paper is a very interesting study of the preparation of ceramic material using microwaves. The use of microwave radiation obviously intensifies and shortens the sintering process. The article is clearly and concisely written.

The following queries and remarks arose during the reviewing process:

  1. In the paragraph dealing with EDS, the authors wrote:

„By comparing EDS results between C-2 and M-2 samples, it is determined that the amount of Fe and Cr in the microwave sample is higher. The same trend was observed in C-3 and M-3 samples.“

How is this possible when, in both cases, glasses with the same initial composition are heated? It appears that the oxygen concentration changed under the action of microwaves, which, when recalculated, could cause a change in the amount of both metals. Do the authors have an idea of ​​what could have happened to the samples composition when exposed to microwaves at the temperatures used?

  1. In the section on Raman spectra, the authors write:

„… and the v of microwave samples is larger than that of conventional samples.“

According to my opinion it does not correspond with the Raman spectra in Fig. 5., the spectra and also their deconvoluted parts look almost the same. The difference does not seem significant.

Author Response

Thank you for your affirmation of the work, and put forward the deficiencies in the article. We have answered your questions and revised them in the manuscipt, as follows:

A1:  Because of the unclear expression in the text, there are differences in understanding. The text has now been modified. The composition of the mother phase glass used in this article is determined and cannot be changed. However, different heat treatment processes will cause differences in the crystal phase composition and structure during the crystallization process. Here, EDS is used to analyze the crystal phase diopside of the sample to obtain the Fe and Cr content changes in the test results.

Action: By comparing EDS results of diopside phase between C-2 and M-2 in glass-ceramic, it is determined that the amount of Fe and Cr in the microwave sample is higher.

A2: It is weak that their deconvoluted peak of Raman spectroscopy, due to the similar component and test environment. Therefore, the results of Raman are counted in Table 3.

Action: Therefore, the ν of the glass-ceramics with the same composition is greater than the ν of the annealed glass samples, and the ν of the microwave heat-treated sample is the largest.

Reviewer 3 Report

The manuscript is devoted to the microwave heating of CMAS system and its crystallization. A set of experimental methods methods is used to characterize the samples. The approach used is interesting and could attract the audience of the journal. However there are few questions to be addressed before possible publication.

  1. DSC methodology is not described and the abbreviation would better be explained.
  2. Coordination number notation Qn should be properly introduced in the text, otherwise it appears suddenly and disappointing.
  3. What is the difference between the three samples in series (for example, C-1, C-2, C-3) in Table 2.
  4. Why the microwave heated samples have different elemental composition, e.g. percentage of chromium? Microwaves could not add any atoms. Was it introduced during synthesis o reflects the composition of initial raw slag?
  5. Decomposition of the broad amorphous Raman spectra into 5+ components and then making conclusions based on this decomposition is a bit fuzzy. Are the authors sure they make no error in this regard. Meanwhile the figure with Raman spectra is overloaded with data and may be separated or truncated.

Author Response

Thank you for your affirmation of the work, and put forward the deficiencies in the article. We have answered your questions and revised them in the manuscipt, as follows:

Action1: DSC is used to determine the thermodynamic properties of the glass. The crystallizing temperature of the glass obtained by water quenching glass at a heating rate of 10°C/min at 1000°C is used as a reference for the glass crystallization heat treatment system

A2: Supplementary explanations are provided in relevant positions in the paper. Qn represents the number of O atoms connected to Si ions in silicate glass.

Action3: The corresponding position in the text has been modified. The samples with different nucleating agent content S1, S2, S3 were heated by muffle furnace and microwave oven for crystallization heat treatment. The samples were named C-1, C-2, C-3 and M-1, M-2, M-3.

A4: Here, the crystalline phase of diopside in the sample is quantitatively analyzed by EDS. The element fixation in the parent glass cannot be changed, but different heat treatment processes can adjust the crystal phase composition during the glass crystallization process. Therefore, different heat treatment processes change the content of Cr in the glass.

A5: The purpose of this paper is to study the influence of different heat treatment methods on the structure of glass-ceramics. Therefore, it is simplified in the process of Raman analysis, and a large number of references show that this simplified treatment is reasonable.

Reviewer 4 Report

The author demonstrated the enhanced crystallization in CaO-MgO-Al2O3-SiO2 (CMAS) glass-ceramics by the microwave sintering.

They suggested the Fe2O3 and Cr2O3 in CMAS play a pivotal role in the diffusion and nucleation by the Raman spectroscopy analysis.

The methods and results should be very interesting for the audience of materials.

There are points to revise:

P4, line 152; Figure 2 should be placed in line 151.
P8, line 245; Please check the subscripts. ν1 (subscript) or ν1? A4% should be A4(subscript)%.
P8, line 254-259; the equations and numbers should be placed in the same line.

I think however that there are a few improvements in the correction of typographical errors and the figure position that should be made before publication.

Author Response

Thank you for your affirmation of the work, and put forward the deficiencies in the article. We have answered your questions and revised them in the manuscipt.
